# A CNN-LSTM Architecture for Detection of Intracranial Hemorrhage on CT scans

**Nhan T. Nguyen**\*                    V.NHANNT64@VINTECH.NET.VN
**Dat Q. Tran**\*                    V.DATTQ13@VINTECH.NET.VN
**Nghia T. Nguyen**                    V.NGHIANT23@VINTECH.NET.VN
**Ha Q. Nguyen**                    V.HANQ3@VINTECH.NET.VN
*Vingroup Big Data Institute, 458 Minh Khai, Hai Ba Trung, Hanoi, Vietnam*

## Abstract

We propose a novel method that combines a convolutional neural network (CNN) with a long short-term memory (LSTM) mechanism for accurate prediction of intracranial hemorrhage on computed tomography (CT) scans. The CNN plays the role of a slice-wise feature extractor while the LSTM is responsible for linking the features across slices. The whole architecture is trained end-to-end with input being an RGB-like image formed by stacking 3 different viewing windows of a single slice. We validate the method on the recent RSNA Intracranial Hemorrhage Detection challenge and on the CQ500 dataset. For the RSNA challenge, our best single model achieves a weighted log loss of 0.0522 on the leaderboard, which is comparable to the top 3% performances, almost all of which make use of ensemble learning. Importantly, our method generalizes very well: the model trained on the RSNA dataset significantly outperforms the 2D model, which does not take into account the relationship between slices, on CQ500. Our codes and models is publicly avaiable at https://github.com/VinBDI-MedicalImagingTeam/midl2020-cnnlstm-ich.

**Keywords:** Computed Tomography, Intracranial Hemorrhage, CNN, LSTM.

## 1. Introduction

The detection of intracranial hemorrhage (ICH) from CT scans of the head is an important problem in healthcare that has recently been received much interest from both the medicine and machine learning communities (Chilamkurthy et al., 2018; Titano et al., 2018; Arbabshirani et al., 2018; Patel et al., 2019). Especially, the ICH Detection challenge (RSNA, 2019) hold by the Radiological Society of North America (RSNA) on the Kaggle framework has attracted 1345 teams from all over the world. During the competition, RSNA has released a large dataset of over 25,000 CT scans, whose each slice is labeled with 5 specific subtypes of ICH. A similar but much smaller dataset of 500 studies is CQ500 (Chilamkurthy et al., 2018), which can be used as a validation set.

The main difficulty of dealing with the RSNA dataset is the 3D representation of a CT scan, which is a stack of 2D images (or slices). This data representation poses many challenges to transfer deep learning techniques on natural images like ImageNet (Deng et al., 2009) to medical imaging tasks, where labeled data is scarce and hard to obtain. A naïve approach to this problem is to ignore the 3D contextual information and to treat

---

\* Contributed equally

every image independently (Chilamkurthy et al., 2018). Alternatively, some studies (Titano et al., 2018; Arbabshirani et al., 2018) have utilized 3D convolutions to learn directly from voxels; however, this approach is computationally expensive and requires training models from scratch on large-scale datasets.

This work proposes a more efficient training strategy for the ICH classification task. Our method attaches a long short-term memory (LSTM) architecture (Hochreiter and Schmidhuber, 1997) to a traditional convolutional neural network (CNN) such that the whole model can be trained end-to-end. The input to the CNN is an RGB-like image obtained by stacking 3 instances of the same slice over 3 different windows that are popularly used in the diagnosis of brain CT. The goal is to take advantage of ImageNet pretrained models while still modeling the spatial dependencies between adjacent slices in 3D space. This approach is similar to that of (Patel et al., 2019), but our work is different in the following aspects:

- Unlike their simple CNN architecture that has to be trained from scratch, we utilize more sophisticated pretrained models such as ResNet (He et al., 2016) and SE-ResNeXt (Hu et al., 2018), which were proven to be powerful on natural images;

- By taking RGB-like images as input, we can adopt various training configurations (He et al., 2019) that have been efficiently used on ImageNet;

- We train and validate our model on public datasets like RSNA and CQ500, while the model in (Patel et al., 2019) was trained and validated on a private dataset.

## 2. Model architecture

For each slice of the CT scan, we use brain window ($l = 40, w = 80$), subdural window ($l = 75, w = 215$), and bony window ($l = 600, w = 2800$) and stack them as three RGB channels. Then, each RGB-like image is fed into a 2D ImageNet-pretrained CNN followed by an LSTM to predict the presence of ICH and its 5 subtypes. The CNN produces 2048 features for each input slice. The LSTM simply consists of two bi-directional LSTM layers with 512 hidden units and one fully connected layer to output sigmoid probabilities of 6 classes at slice level. The overall architecture is sketched in Fig. 1

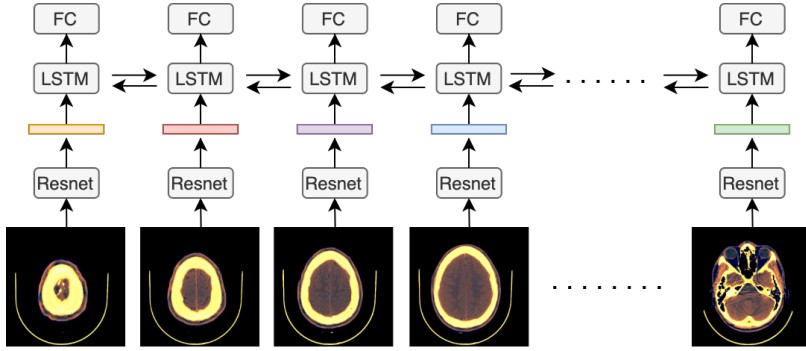

Figure 1: The overall CNN-LSTM architecture for the ICH classification task.

## 3. Experiments

### 3.1. Datasets and training procedure

The RSNA dataset comprised of over 25,000 non-contrast brain CT scans, each of which has between 20 to 60 slices of size 512×512. The labeling was manually performed on each slice with 5 subtypes of ICH: intraparenchymal, intraventricular, subarachnoid, subdural and epidural. For the purpose of the ICH Detection challenge on Kaggle, the RSNA dataset was split into a public train, a public test, and a private test. We used exactly these sets for the training, validation, and testing of our algorithm, respectively. The models trained on RSNA were further tested on 491 studies of the CQ500 dataset; the other 9 studies are outliers and so excluded from the dataset. Each study in this dataset has between 16 to 128 slices and was annotated similarly to the RSNA dataset.

We experimented with two ImageNet-pretrained CNNs: ResNet-50 and SE-ResNeXt-50. The overall architecture was then trained solely on RSNA. To that end, we applied the bag of tricks proposed in (He et al., 2019) with some modifications. All models were trained for 30 epochs using the Adam optimizer with an initial learning rate of $1e-3$ and cosine annealing schedule with linear warm-up. During training, we performed various augmentations: random cropping and resizing, horizontal/vertical flip, random rotation between 0° and 30°, optical distortion, grid distortion, Gaussian noise, and CutMix (Yun et al., 2019) with $\alpha = 1.0$. The validation set was used to checkpoint the best model weights.

### 3.2. Results

The performances of our two single models on RSNA, shown in Table 1, are measured by the weighted log loss. Specifically, the overall loss is computed by averaging the 6 binary cross-entropy losses on the 5 ICH subtypes and the additional class of any ICH with the corresponding set of weights $(\frac{1}{7}, \frac{1}{7}, \frac{1}{7}, \frac{1}{7}, \frac{1}{7}, \frac{2}{7})$. Our single-model results are on par with top 3% rankings on the Kaggle leaderboard, where model ensembles are allowed.

When validating on CQ500, we only took the maximum predicted probabilities of all slices as the scan-level result and compared the performance to the random-forest-based method proposed by `Qure.ai` in (Chilamkurthy et al., 2018). Table 2 reports the classification performances in terms of AUC (Area Under Curve) of our 2 models on CQ500 in comparison with the original method. It is noteworthy that despite the data distribution shift and not being optimized for the scan-level prediction task, our models still generalize very well and outperform that of (Chilamkurthy et al., 2018) by large margins.

| Models | Weighted Log Loss |
|---|---|
| ResNet-50 | 0.05289 |
| SE-ResNeXt-50 | 0.05218 |

Table 1: Performance on the private test set of the RSNA ICH Detection challenge.

## References

M.R. Arbabshirani, B.K. Fornwalt, G.J. Mongelluzzo, J.D. Suever, B.D. Geise, A.A. Patel, and G.J. Moore. Advanced machine learning in action: Identification of intracranial

| | AUC | | |
|---|---|---|---|
| Finding | `Qure.ai` | ResNet-50 | SE-ResNeXt-50 |
| Intracranial Hemorrhage | 0.9419 | 0.9597 | **0.9613** |
| Intraparenchymal | 0.9544 | 0.9616 | **0.9674** |
| Intraventricular | 0.9310 | **0.9901** | 0.9858 |
| Subarachnoid | 0.9574 | 0.9662 | **0.9696** |
| Subdural | 0.9521 | **0.9654** | 0.9644 |
| Extradural (Epidural) | 0.9731 | **0.9740** | 0.9731 |

Table 2: Performance on CQ500 in comparison with the method of Qure.ai.

hemorrhage on computed tomography scans of the head with clinical workflow integration. *npj Digital Medicine*, 1:1–7, 2018.

S. Chilamkurthy, R. Ghosh, S. Tanamala, M. Biviji, N.G. Campeau, V.K. Venugopal, V. Mahajan, P. Rao, and P. Warier. Development and validation of deep learning algorithms for detection of critical findings in head CT scans. *Lancet*, 392(10162):2388–2396, 2018.

J. Deng, W. Dong, R. Socher, L.-J. Li, K. Li, and L. Fei-Fei. Imagenet: A large-scale hierarchical image database. In *CVPR*, pages 248–255, 2009.

K. He, X. Zhang, S. Ren, and J. Sun. Deep residual learning for image recognition. In *CVPR*, pages 770–778, 2016.

T. He, Z. Zhang, H. Zhang, Z. Zhang, J. Xie, and M. Li. Bag of tricks for image classification with convolutional neural networks. In *CVPR*, pages 558–567, 2019.

S. Hochreiter and J. Schmidhuber. Long short-term memory. *Neural Computation*, 9(8): 1735–1780, 1997.

J. Hu, L. Shen, and G. Sun. Squeeze-and-excitation networks. In *CVPR*, pages 7132–7141, 2018.

A. Patel, S. C. van de Leemput, M. Prokop, B. Van Ginneken, and R. Manniesing. Image level training and prediction: Intracranial hemorrhage identification in 3D non-contrast CT. *IEEE Access*, 7:2169–3536, 2019.

RSNA. Intracranial hemorrhage detection challenge. https://www.kaggle.com/c/rsna-intracranial-hemorrhage-detection/, 2019.

J.J. Titano, M. Badgeley, J. Schefflein, M. Pain, A. Su, M. Cai, N. Swinburne, J. Zech, J. Kim, J. Bederson, J. Mocco, B. Drayer, J. Lehar, S. Cho, A. Costa, and E.K. Oermann. Automated deep-neural-network surveillance of cranial images for acute neurologic events. *Nature Medicine*, 24:1337–1341, 2018.

S. Yun, D. Han, S.J. Oh, S. Chun, J. Choe, and Y. Yoo. CutMix: Regularization strategy to train strong classifiers with localizable features. In *ICCV*, pages 6023–6032, 2019.

