# OpenReview forum: "A CNN-LSTM Architecture for Detection of Intracranial Hemorrhage on CT scans"
_MIDL.io/2020/Conference — MIDL 2020_

### Official Review · AnonReviewer3 · 2020-03-06
**An unsurprising solution to a problem that has many other solutions of equal or better performance**

**Rating:** 2
**Confidence:** 5

**Review:**

The authors suggest to use a combination of transfer learning from pretrained CNNs with bidirectional LSTM layers to perform 3D CT volume classification in data where they assume that a pre-trained classification network should be useful. They compare their results with the leaderboard of the RSNA 2019 ICH classification challenge, where one of the two used datasets was acquired, and with the priorly reported results on the Qure.ai public head CT dataset. While in the RSNA challenge, they score somewhere between 30th and 40th position in the leaderboard, and they outperform Qure.AI by a  small but consistent margin.

The paper is clearly written with only few typos, concisely short, and overall easy to read. The authors cite relevant work that is close to their approach, and try to explain why and how theirs differs.

I don't fully agree with the points they make, however. In particular using RGB-like images composed of differently windowed DICOM images makes little sense in general. Trainable convolutional layers prepended to the pre-trained network will likely do a better job in preprocessing the gray scale images than hand-selected window/level values, which are completely tailored to the human eye and a clinical task, not a computer eye. Also, the fact that the model is validated on a public dataset is no strong point. The authors also point out that their work is no ensemble classifier -- but they don't explain why this is a benefit. Likewise, the efficiency of training is also only interesting with the limited resources available in Kaggle, but not "in the wild", where training any native 3D network, even from scratch, can and should surely be conducted, if superior results can be expected.

More importantly, though, I find the basic motivation of the paper ("there are no pre-trained 3D classifiers I could use, but I want to participate in the RSNA challenge") too weak to accept it. In the light of the challenge, with limited time and limited resources, it is certainly acceptable to go with such an (unsurprising) approach. The solution apparently works well, and can be applied practically to similar whole-volume classification tasks with weak labels (i.e., those with no spatial clues). On the other hand, the motivation is not given by a clinical task, but to a certain degree by the setup of the challenge.

Taking into consideration that there are more than 30 better performing solutions in the challenge, and that also the feature-engineering approach of Qure.AI is just a little bit worse, I can't see the practical benefit of the presented work.

Further taking into account that the approach is not novel or surprising, overall I cannot suggest it for publication.

---

### Official Review · AnonReviewer2 · 2020-03-11
**The idea of using three windows of CT image to mimic a RGB image is interesting**

**Rating:** 3
**Confidence:** 4

**Review:**

This paper uses three windows of CT image to mimic a RGB image so that effective models pretrained on ImageNet can be utilized on medical images.  A CNN-LSTM architecture is used to detect intracranial hemorrhage from a series of CT slices.   The paper is clearly presented, and experiments on two datasets show that the proposed method works fairly well.

Stregnth
1. Using three windows of CT image to mimic a RGB image, which make it possible to utilize models trained for natual images on medical images.
2. Experiments on two datasets show that the proposed idea works fairly well, especially the models trained on the bigger dataset generalize well on the smaller dataset.

Weakness:
1. Methodology contribution is minimal.
2. Performance on the ICH Detection challenge fall at the postion about top 8% now (somewhere around 35th).

---

### Official Review · AnonReviewer4 · 2020-03-14
**CNN-LSTM architectures allow (slice-by-slice) CT classification with spatial volumetric consistency**

**Rating:** 4
**Confidence:** 4

**Review:**

Although I disagree with the claim of novelty of this method, I find the paper interesting and appropriate for MIDL presentation. I think this approach has the merit of showing a practical application where the use of LSTMs to incorporate spatial consistency for volumetric problems is appropriate. This idea has been previously used for segmentation and, as far as I know, pre-trained models were not leveraged in that work. Here, the authors propose to re-use imagenet pre-trained models, trained with slice-wise labels, and enforce consistency over volumes through a bidirectional LSTM. This has a series of advantages such as reduced compute resources needed for training and the ability of using pre-trained models from 2D applications, which are very common in computer vision.

The description of datasets and methods is pretty accurate. The method is trained on the RSNA dataset published on Kaggle and tested on the GQ500 dataset. The results are convincing and they reveal the merits of the model. The model can in fact compete with methods making use of ensemble prediction and obtain comparable if not superior results.

Table 1 could be omitted, so that Table 2 would fit within 3 pages.

I would like to accept this paper. My rating is between weak accept (because of the limited novelty) and strong accept.

---

### Official Review · AnonReviewer1 · 2020-03-14
**Simple but seemingly effective approach for large 3D ICH volume data**

**Rating:** 3
**Confidence:** 3

**Review:**

The paper presents a method which combines a commonly used CNN (ResNet and ResNeXt) and LSTM which uses the slices as "sequences". The method is quite simple: the CNN output is being used as the input to an LSTM, so it is difficult to give credit to the model formulation novelty. However, it appears to outperform an existing method, but the comparing method is a random-forest-based method. While I understand that the authors avoided unfair comparisons against their method to other ensemble-based methods, I am not sure if the presented comparison against a RF-based method is a fair one. The overall paper was well written and easy to understand.

---

### Meta-Review · Area_Chair1 · 2020-04-07
**MetaReview of Paper41 by AreaChair1**

**Rating:** 3

**Metareview:**

While there are some concerns regarding the limited novelty of the method, the majority of the reviewers find this paper of sufficient merit to justify acceptance.

**Paper Type:**

validation/application paper

---

### Decision · Program_Chairs · 2020-04-11

Accept